# Polyprev: Randomized, Multicenter, Controlled Trial Comparing Fecal Immunochemical Test with Endoscopic Surveillance after Advanced Adenoma Resection in Colorectal Cancer Screening Programs: A Study Protocol

**DOI:** 10.3390/diagnostics11091520

**Published:** 2021-08-24

**Authors:** Cristina Regueiro, Raquel Almazán, Isabel Portillo, María Besó, Carlos Tourne-Garcia, Elena Rodríguez-Camacho, Akiko Ono, Ángel Gómez-Amorín, Joaquín Cubiella

**Affiliations:** 1Department of Gastroenterology, Instituto de Investigación Sanitaria Galicia Sur, Hospital Universitario de Ourense, 32005 Ourense, Spain; joaquin.cubiella.fernandez@sergas.es; 2Conselleria de Sanidade, Dirección Xeral de Saúde Pública, 15704 Galicia, Spain; RAQUEL.ALMAZAN.ORTEGA@sergas.es (R.A.); Elena.Rodriguez.Camacho@sergas.es (E.R.-C.); Angel.Gomez.Amorin@sergas.es (Á.G.-A.); 3Osakidetza Basque Health Service, Basque Country Colorectal Cancer Screening Programme, 48009 Bilbao, Spain; MARIAISABEL.PORTILLOVILLARES@osakidetza.eus; 4Biocruces Health Research Institute, Cancer Biomarker Area, 48903 Barakaldo, Spain; 5Servicio de Promoción de la Salud y Prevención en el Entorno Sanitario, Dirección General de Salud Pública y Adicciones, 46021 Valencia, Spain; beso_mardel@gva.es; 6Colon and Rectal Cancer Prevention Program, Directorate General for Public Health, Autonomous Government for Health, 30008 Mucia, Spain; carlosi.tourne@carm.es; 7Department of Gastroenterology, Hospital Clínico Universitario Virgen de la Arrixaca, 30120 Murcia, Spain; ono.akiko@gmail.com

**Keywords:** colorectal cancer, screening programs, fecal immunochemical test, endoscopic surveillance

## Abstract

Colorectal cancer (CRC) screening programs have been implemented to reduce the burden of the disease. When an advanced colonic lesion is detected, clinical practice guidelines recommend endoscopic surveillance with different intervals between explorations. Endoscopic surveillance is producing a considerable increase in the number of colonoscopies, with a limited effect on the CRC incidence. Instead, participation in CRC screening programs based on the fecal immunochemical test (FIT) could be a non-inferior alternative to endoscopic surveillance to reduce 10-year CRC incidence. Based on this hypothesis, we have designed a multicenter and randomized clinical trial within the Spanish population CRC screening programs to compare FIT surveillance with endoscopic surveillance. We will include individuals aged from 50 to 65 years with complete colonoscopy and advanced lesions resected within the CRC screening programs. Patients will be randomly allocated to perform an annual FIT and colonoscopy if fecal hemoglobin concentration is ≥10 µg/g, or to perform endoscopic surveillance. On the basis of the non-superior CRC incidence, we will recruit 1894 patients in each arm. The main endpoint is 10-year CRC incidence and the secondary endpoints are diagnostic yield, participation, adverse effects, mortality and cost-effectiveness. Our results may modify the clinical practice after advanced colonic resection in CRC screening programs.

## 1. Introduction

Colorectal cancer (CRC) is one of the most common malignancies in western countries. About 40,000 new cases of CRC are diagnosed each year in Spain, and 39% of those affected die due to this disease [1,2,3]. CRC screening programs have been implemented in order to reduce the burden of the disease. Screening programs are used to detect the CRC in initial stages. Furthermore, during the colonoscopy, CRC screening programs try to identify and resect the precursor lesion of the disease: the colorectal adenoma and the serrated lesions [4]. In this way, screening programs achieve a 30% reduction in the CRC incidence and a 50% reduction in CRC mortality [5].

Screening programs in Spain are based on the biennial detection of fecal hemoglobin (f-Hb) with a fecal immunochemical test (FIT) and a colonoscopy if f-Hb ≥ 20 µg/g of feces [6]. Depending on the characteristics of the resected colonic lesions, patients are classified into two different groups [7]. A low-risk group is defined when 1–2 adenomas are detected (<10 mm, tubular histology and low-grade dysplasia). On the contrary, a high-risk group is defined when at least one advanced adenoma (≥10 mm, villous histology and high grade displasia) or one serrated lesion (≥10 mm and displasia) is detected [8,9,10]. Clinical practice guidelines (CPG) have taken these characteristics into account when determining surveillance recommendations and intervals [3,8,9,10]. Patients with low-risk lesions have been routinely recommended to return to population screening programs, whereas patients with high-risk lesions are recommended to perform endoscopic surveillance with different intervals between explorations [9,11,12]. The implementation of this CRC screening strategy is producing a considerable increase in the number of colonoscopies, with a significant cost to the health system, and consuming a high proportion of the colonoscopy capacity [6]. The rate of progression of advanced adenoma to CRC is estimated to be low (2.6% in population aged 50 to 59 years and 5.6% in the population ≥80 years). Remarkably, endoscopic surveillance reduces CRC mortality by only 1.7% and increase the number of colonoscopies by 62% [13]. Moreover, colonoscopy is a procedure associated with serious side effects [4].

A recently conducted British study has evaluated the FIT diagnostic yield for CRC and advanced adenomas detection after resection of advanced lesions. With a highly sensitive cut-off (10 µg/g of feces), the sensitivity and specificity for CRC was 84.6% and 70.8% respectively, with a significant cost reduction (£2633 vs. £956 per patient) [14]. Furthermore, other studies show that the British population prefer to perform non-invasive fecal test rather than colonoscopy (60.8% vs. 31%) [15]. Based on this evidence, we have designed a randomized clinical trial within the population CRC screening programs, comparing FIT surveillance to endoscopic surveillance in patients after advanced lesion resection.

## 2. Materials and Methods

### 2.1. Hypothesis

After the resection of high-risk adenomas detected within the CRC screening program, the 10-year CRC incidence in patients participating in a CRC screening programs based on annual FIT determination is not superior to the incidence in patients undergoing endoscopic surveillance.

### 2.2. Objectives

The main objective of our study is to compare the 10-year CRC incidence between patients undergoing endoscopic surveillance and patients participating in a population CRC screening program based on annual FIT determination. Additionally, we have defined the following as secondary objectives:
-To compare the diagnostic yield for CRC and advanced adenomas detection between both groups at a 3 year interval.-To evaluate the effect of the two surveillance strategies on the following variables: mortality (global and associated with CRC), colonic lesion detection, participation in the surveillance strategy, adverse effects, use of health resources and cost-effectiveness.-To determine the values and preferences of subjects with advanced adenomas resected endoscopically regarding the type of surveillance and the risk of CRC.

### 2.3. Study Design

We designed this study as a randomized, controlled and multicenter clinical trial. We will include patients from the CRC screening programs of Galicia, the Basque Country, the Region of Valencia and Murcia. This study has been registered in ClinicalTrials.gov (ID: NCT04967183).

### 2.4. Inclusion Criteria

Individuals aged from 50 to 65 years with at least one advanced adenoma (tubulovillous or villous histology, high-grade dysplasia or ≥10 mm), and/or at least three non-advanced adenomas, detected and resected completely within the population-based CRC screening program.

### 2.5. Exclusion Criteria

Individuals with (1) personal history of CRC, (2) colonic lesion ≥ 10 mm resected without histological diagnosis, (3) more than 10 adenomas in baseline colonoscopy, (4) serrated polyposis syndrome, (5) two or more first-degree relatives with CRC, (6) hereditary predisposition to CRC, (7) relevant comorbidity with life expectancy inferior to 5 years, (8) colonoscopy with incomplete mucosal examination (no cecal intubation, Boston Score < 6 or <2 in any of the sections), (9) incomplete resection of baseline lesions. (10) piecemeal resection of sessile of flat lesions ≥20 mm and (11) non-acceptance after reading the informed consent.

### 2.6. Study Development: Inclusion

First, patients will have a telephone interview to confirm they meet the inclusion criteria and do not meet any exclusion criteria. Then, we will invite the patients to participate in the study after explaining the trial. In the case of acceptance, we will send them the documentation, including an explanatory brochure, a values and preferences survey, the information sheet and the informed consent that will be returned by postal mail. After receiving the signed informed consent, the patient will be randomly allocated to perform an annual FIT and colonoscopy if fecal hemoglobin concentration is ≥10 µg/g, or to perform endoscopic surveillance. Patients will be also stratified into two groups: ≥5 adenomas or an adenoma ≥20 mm, and the remaining patients. Finally, patients will be informed of the group to which they were assigned by a postal letter, including the recommendation for follow-up. The protocol includes a follow-up period of 10 years.

### 2.7. Study Development: Group I

The CRC screening program will send a kit annually to collect a fecal sample until the surveillance period is completed (10 years) (Figure 1). We will set a cut-off of ≥10 µg hb/g of feces to refer to colonoscopy, which will be performed within the CRC screening program agendas. After performing a work-up colonoscopy, FIT will be sent to the patient:
-After one year if the colonoscopy was incomplete or a lesion requiring endoscopic surveillance was completely resected.-After five years if the colonoscopy evaluated the entire mucosa, was normal or had lesions that do not require endoscopic surveillance (1–2 non-advanced adenomas).

### 2.8. Study Development: Group II

The first surveillance colonoscopy will be performed in a three-year interval (Figure 1). If an advanced adenoma or at least three non-advanced adenomas are detected, colonoscopy will be repeated after three years. On the contrary, if colonoscopy is normal or 1–2 non-advanced adenomas are detected, colonoscopy will be repeated after five years. Surveillance colonoscopy will be considered as the one performed in the interval of 6 months before and 6 months after the established surveillance date. The remaining colonoscopies will be considered unscheduled. Surveillance colonoscopies will not be performed if an unscheduled colonoscopy was performed recently, one of the study endpoints is reached, or there is comorbidity that contraindicates the colonoscopy or the patient rejects.

### 2.9. Sample Size Calculation

We designed the study on the basis that CRC incidence in the two groups will be 1.24% [16], with a 1:1 ratio between both groups and a non-superiority limit in 1% [17]. Assuming a 0.8 B error and an alpha error of 0.05, 1515 patients should be included in each group. Accounting for an estimated dropout percentage of 20%, it is necessary to recruit 1894 patients in each group, leading to a total of 3788 subjects in the study. This sample size calculation was carried out with the statistical program Ene 3.0.

### 2.10. Endpoints


-Invasive CRC: this is the main endpoint of the study and is defined as a colonic adenocarcinoma that invades the submucosa. We will not consider adenocarcinomas in situ and intramucosal carcinomas as invasive CRC.-Interval CRC: this is defined as the CRC detected between two organized surveillance (FIT or colonoscopy).-Mortality: we will collect the deaths and their cause (associated with CRC, associated with adverse effects or unrelated).-Colonic lesions: we will classify colonic lesions as advanced or not advanced. We will define advanced lesions as advanced adenomas (some of at least 10 mm, hairy histology or high grade dysplasia) or advanced serrated lesions (some of at least 10 mm or with dysplasia). The rest will be classified as not advanced.-Participation in the surveillance strategy: we will define three categories to evaluate the participation: non-participation, irregular and regular participation.-Adverse effects: adverse effects associated with surveillance are defined as complications that require hospitalization. Those related to the surgical treatment of benign colonic lesions will be also included as adverse effects.


### 2.11. Exit from the Study

Patients will leave the study due to death, CRC diagnosis, withdrawal of consent to participate or completion of the study period. At the end of the 10 years, patients assigned to group I will remain in the population-based CRC screening program. Patients in group II will continue to perform endoscopic surveillance according to the recommendations established by the existing CPG.

### 2.12. Data Management

We will collect all the information in a database designed for this purpose. Regarding baseline data, we will include the inclusion and exclusion criteria, demographic information and the characteristics of the initially resected colonoscopies and polyps. We will also collect the questionnaires administered, the intervention information, the FIT rounds, all the colonoscopies performed and the appearance of the study endpoints in the database. The study requires the long-term follow-up of the patient, so it is possible that they could change their address. In this case, a pseudo-anonymized identification number will be used to determine the appearance of CRC and mortality, in accordance with the general data protection regulation (2016/679) and the Law on Protection of Personal Data and guarantee of digital rights (Organic Law 3/2018).

### 2.13. Study Monitoring

An independent monitoring and security committee will be created externally to review the progress of the study and ensure it is carried out according to the ethical rules. The committee members will be appointed by the principal investigators and will not have any role in the progression of the study. Additionally, internal rules will be established in the database to identify inconsistencies in the information included. A calendar of audits of the archive information will be also scheduled in each of the centers.

### 2.14. Ethical and Legal Aspects

The study was designed according to the Declaration of Helsinki, in the Council of Europe Convention on Human Rights and Biomedicine, and according to the Spanish legislation in the field of biomedical research, the protection of personal data and bioethics. Specifically, the management of both the data collected and the medical records will comply at all times with the requirements of the general data protection regulation (2016/679) and Organic Law 3/2018 of Protection of Personal Data and guarantee of digital rights. Ethical approval was obtained from the Clinical Research Ethics Committee of Galicia, Spain (2020/053). Informed consent will be obtained from all study participants. Any possible protocol modification will be communicated to the ethical committee and to all relevant parties.

### 2.15. Statistical Analysis

The primary analytical approach to the trial will follow the principle of intention to treat (ITT). We will compare the differences for each endpoint between the groups by calculating the incidence, incidence density, risk differences and applying the log-rank test. We will also perform an analysis per protocol to estimate the causal effect that would have been observed if all the people in the intervention arms had met.

For the analysis of the diagnostic yield at three years, we will compare the rates of CRC, adenoma and advanced serrated lesions detection using the risk ratio and relative risk. Logistic regression will be performed to adjust the risk ratio using age, sex, screening program and stratification as confusing variables.

We will also develop a cost–utility Markov model to evaluate the cost-effectiveness of CRC screening. The model will consider different health states based on the usual clinical progression of CRC. We will use the data obtained from this clinical trial, such as transition probabilities, sensitivity, specificity, incidence and mortality rates due to CRC. The EuroQol questionnaire (EQ–5D) [18] will be used to measure the utilities associated with the CRC.

## 3. Discussion

This trial was designed to evaluate the potential of annual FIT for CRC surveillance of patients with advanced colonic lesions instead of surveillance colonoscopy. We will evaluate a different screening strategy based on an annual FIT test using a low fecal hemoglobin threshold and colonoscopy in positive cases. We selected a 10-year period according to previous recommendations for surveillance after adenoma resection [19]. Additionally, the main goal of our study is the prevention of the disease, which is why a 10-year period is adequate to evaluate CRC incidence. On the other hand, we think the CRC incidence in the FIT based strategy is non-superior to colonoscopy surveillance and in order to establish the 1% margin, we used the magnitude of benefit that typical members of the population would value to opt for screening [17]. Additionally, we chose a superior margin to other ongoing trials, because a superior CRC incidence would be outweighed by a reduction in the cost of surveillance, the number of colonoscopies and the rate of associated complications [20].

The diagnostic accuracy of FIT is high in terms of the CRC detection rate in CRC screening programs [21]. Hence, its accuracy can reach 100% sensitivity and 90–94% specificity for CRC detection in CRC-screening programs and evaluation of symptomatic patients [22]. On the contrary, FIT has a limited diagnostic accuracy for advanced colonic lesion at a single determination. In asymptomatic Spanish subjects recruited at the hospitals included in our study, a single FIT determination detects 31% and 21% of advanced adenomas at the 10 µg Hb/g and 20 µg Hb/g of faeces thresholds, respectively, with a specificity higher than 90%. There are several characteristics of the adenomas associated with a positive FIT: number, location, morphology and size [21]. However, the strength of an FIT-based CRC screening is that it is based on periodic (annual or biennial) determination [23]. A recent British study shows that the sensitivity for CRC and advanced adenomas increases with each annual FIT round [14]. Furthermore, the threshold used can be tailored according to colonoscopy capacity and long-term objectives. However, not all the adenomas will progress to a CRC, and the aim of a surveillance strategy is to detect only the adenoma that will progress to CRC in the next surveillance period.

On the other hand, the evidence regarding the benefits of endoscopic surveillance in high-risk lesions is limited to cohort studies. In a study published by Cottet et al., in 2012 [24], the standardized incidence ratio (SIR) was 1.10 (95% CI: 0.62–1.82) and 4.26 (95% CI: 2.89–6.04) in those patients with and without colonoscopy follow-up, respectively. Atkin et al., also showed, in a study including 12,000 patients with high-risk lesions (1–2 adenomas ≥10 mm or 3–4 adenomas <10 mm), that performing at least one endoscopic surveillance reduces the incidence of CRC (HR 0.57, 95% CI: 0.40–0.80). However, this risk reduction was limited to a subgroup of patients: low-quality colonoscopy, large (≥20 mm), high-grade dysplasia and proximal adenomas [12]. In this respect, the available guidelines recommend performing baseline colonoscopy with a full exploration of the colonic mucosa and resection of all detected polyps [9,11,12]. CRC detection during surveillance depends not only on the characteristics of the polyps but also on the endoscopist’s technical ability. A recently published Polish study reveals that long-term risk of CRC is increased (HR 2.69, 95% CI: 1.62–4.47) if baseline colonoscopy is performed by low-performing endoscopists (adenoma detection rate <20%) [25]. In this sense, Spanish CRC screening programs initiate surveillance after a complete evaluation of colonic mucosa and resection of colonic lesions. Furthermore, they perform a continuous control of the endoscopist quality indicators [26]. The increased quality of the baseline exploration produces a stage migration. On one hand, the number of patients for which we recommend endoscopic surveillance is increasing. On the other hand, the most recently published articles showed a lower-than-expected long-term CRC incidence. This is an additional justification for our clinical trial [11].

The potential use of annual FIT in CRC surveillance can be also beneficial, as it can reduce the risk of complications associated with colonoscopy. Some of these effects are intestinal perforation, gastrointestinal hemorrhage after resection of colonic lesions and systematic complications associated with the administered sedation [4]. Additionally, the risk effects related to colonoscopy can also cause discomfort for patients. In fact, a British study shows that most of the population prefer FIT to colonoscopy for a surveillance strategy [15]. We also think that annual FIT may be a more attractive strategy than colonoscopy and, consequently, a means of increasing participation rates in CRC surveillance.

The economic cost associated with colonoscopies is another disadvantage when compared with participation in an FIT-based CRC screening. Colonoscopy surveillance is expensive and increases the number of colonoscopies, with an additional cost of €68,000 for an increase of 0.9 years of life [13]. This problem can be also solved by annual FIT surveillance [14]. Surveillance based on annual FIT would reduce the number of colonoscopies with the associated reduction in economic cost. In the British trial, the number of patients that required work-up colonoscopy when using the 10 μg Hb/g threshold was 28.8% [14]. The incremental cost-effectiveness of colonoscopy versus FIT surveillance was £7354 per additional advanced adenoma detected, and £180,778 per additional CRC detected [14].

We limited our analysis to patients aged from 50 to 65 years. Although this could limit the transference of our final results, it is necessary to obtain a 10-year period of surveillance within the CRC screening programs. Subjects are kept under surveillance in the screening programs in Spain until 75 years. Furthermore, the available practice guidelines recommend surveillance in patients between 75 and 80 years old only if they maintain an excellent performance status with minimum comorbidity [9,10,11].

In sum, we think that annual FIT, using a low fecal hemoglobin cut-off and colonoscopy in positive cases, can reach a high diagnostic yield, could be accepted by most patients and would be cost-saving compared with 3-yearly colonoscopy. Following this, annual FIT could be a new potential strategy in CRC surveillance for patients, after resection of advanced colonic lesions.

## Figures and Tables

**Figure 1 diagnostics-11-01520-f001:**
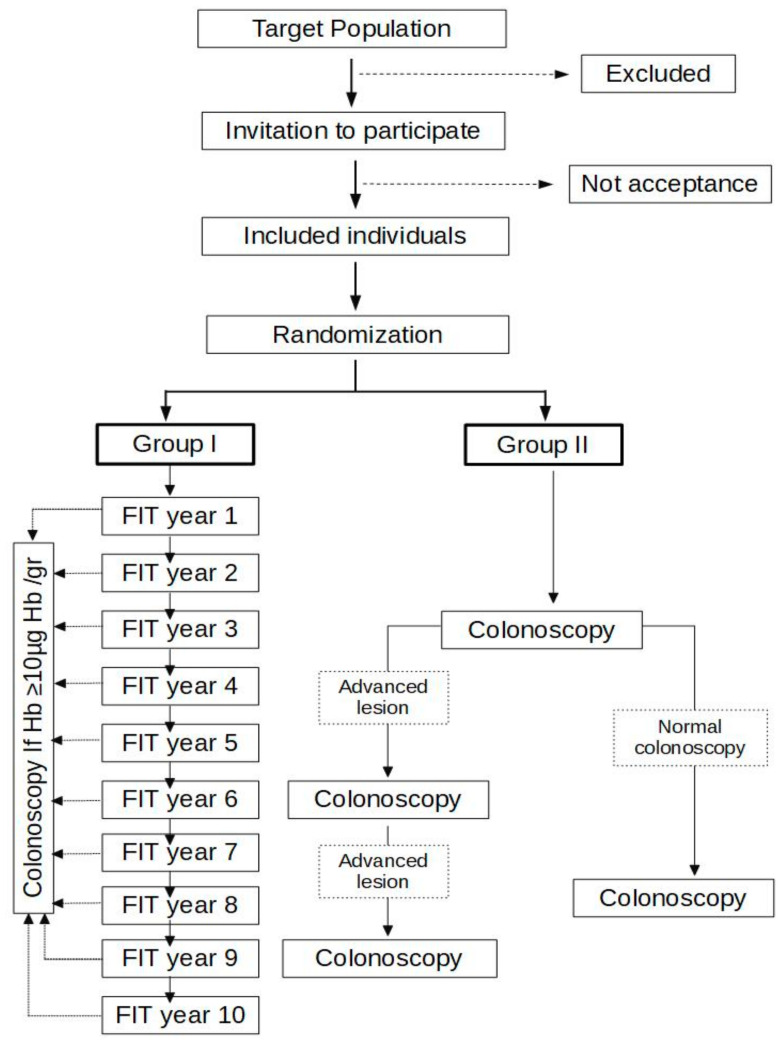
Diagram of the study design.

## Data Availability

Participant files and data will be stored in a secure and accessible manner for the period set by the current guidelines after completion of the study.

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
