# Peer review of "Polyprev: Randomized, Multicenter, Controlled Trial Comparing Fecal Immunochemical Test with Endoscopic Surveillance after Advanced Adenoma Resection in Colorectal Cancer Screening Programs: A Study Protocol"

_diagnostics, 2021, doi:10.3390/diagnostics11091520_

Round 1

Reviewer 1 Report

The Authors describe the plan for a very  ambitious study aimed  compare in a randomized prospective study FIT versus periodic endoscopies after endoscopic resection of advanced adenomas. Results will be analyzed determining by colonoscopy the prevalence of missed CRC after 10 years.

The hypothesis is that participation in CRC screening programs based on the fecal immunochemical  test (FIT) could be a non-inferior alternative to endoscopic surveillance to reduce 10 year CRC  incidence. Based on this hypothesis, the Authors  have designed a multicenter and randomized clinical trial within the Spanish population CRC screening programs to compare FIT surveillance with  endoscopic surveillance. The Authors  will include individuals aged 50 to 65 years with complete colonoscopy  and advanced lesions resected within the CRC screening programs. Patients will be randomly allocated to perform an annual FIT and colonoscopy if fecal hemoglobin concentration is ≥10μg/g,  or to perform endoscopic surveillance. On the basis of the non-superior CRC incidence, 1,894 patients will be recruited bin each arm. The main endpoint is 10 year CRC incidence and the secondary endpoints are diagnostic yield, participation, adverse effects, mortality and cost-effectiveness

POSITIVE POINTS

1-FIT screening leads to significant cost reduction.  General  population prefer to perform non-invasive  fecal test rather than colonoscopy  .

2- According to the Authors,  the rate of progression of advanced adenoma to CRC is estimated to be low (2.6 % in population aged 50 to 59 years and 5.6 % in the population ≥ 80 years). Remarkably, endoscopic surveillance reduces CRC mortality only 1.7% and increase the number of colonoscopies by 62 % . Moreover, colonoscopy is a procedure  associated with serious side effects .

CRITICISMS

1-The Authors have hypothesized the role of FIT in the screening program based on data from British population. I would advice, if I might, to consider to perform a pilot study to determine the accuracy, sensitivity and sensibility of FIT and the optimal diagnostic cut-off of Hb in the feces in the Spanish centers where the trial will be carried on.

2-The Authors will analyze data after 10 years. This is the time usually reported as interval in the adenoma-carcinoma sequence in sporadic CRC. Probably, it might be wiser to perform in all patients a colonoscopy five year after entering the study to have  a more accurate assessment avoiding the risk of missing lesions in the FIT arm.

3-The Authors will include only patients aged from 50 to 65 years of age. However, in all screening programs included patients are usually older than 65; CRC is diagnosed more often with advanced age. Being the life expectancy in Spain is around 80 years in males and 83 in females, it could be important to include also older patients.

4-A very detailed Informed Consent From is required to explain the detail of the study to the patients.

5-The Authors stated that the general population prefer to have  performed FIT rather than  colonoscopy. Is this true also for patients who had a diagnosis of advanced adenoma?  These patients probably were told about the risk they overcame having the adenoma resected.  Are they ready to risk trying a new diagnostic test? I do not have an answer. Do the Author have an answer?

OTHER SUGGESTIONS

The study is very ambitious and it will require significant efforts. I would advice to include in the analysis a study about the use or not of anti-inflammatory drugs (aspirin,statins) which seem to have an important role in preventing occurrence and progression of CRC adenomas.

Author Response

1-The Authors have hypothesized the role of FIT in the screening program based on data from British population. I would advice, if I might, to consider to perform a pilot study to determine the accuracy, sensitivity and sensibility of FIT and the optimal diagnostic cut-off of Hb in the feces in the Spanish centers where the trial will be carried on.

We have not only based our hypothesis on data from British population. Cubiella J et al [1] evaluated the FIT diagnostic accuracy using 474 asymptomatic Spanish subjects from the centers included in our trial. Authors considered positive FIT using 2 different cut-off: 10 µg Hb/g and 20 µg Hb/g of faeces. 61 patients showed a positive result at a 20 µg Hb/g threshold whereas 71 patients were positive at 10 µg Hb/g cut-off. Patients with advanced adenomas, distal advanced adenomas, high risk or intermediate risk lesions had more probabilities to show a positive test. Remarkably, authors did not find statistically differences between both threshold. A change has been made in the second paragraph of the discussion (lines 251-252) to clarify we have also considered data based on Spanish population to design our study.

[1[ Cubiella J et al. Characteristics of adenomas detected by fecal immunochemical test in colorectal cancer screening. Cancer Epidemiol Biomarkers Prev. 2014 Sep;23(9):1884-92.

2-The Authors will analyze data after 10 years. This is the time usually reported as interval in the adenoma-carcinoma sequence in sporadic CRC. Probably, it might be wiser to perform in all patients a colonoscopy five year after entering the study to have a more accurate assessment avoiding the risk of missing lesions in the FIT arm.

We have designed our study based on previous recommendations for surveillance after advanced adenomas resection [2]. A 10 year period is recommended due to CRC prevention is the main objective of this surveillance. The main goal of our study is also the prevention of the disease rather than reduction of mortality by early detection. In this way, we think 10 years is a reasonable period to evaluate CRC incidence after this time. We have modified the first paragraph (lines 238-240) of the discussion to clarify this idea.

[2] Rutter MD at al. Principles for Evaluation of Surveillance After Removal of Colorectal Polyps: Recommendations From the World Endoscopy Organization. Gastroenterology. 2020 May;158(6):1529-1533.e4.

3-The Authors will include only patients aged from 50 to 65 years of age. However, in all screening programs included patients are usually older than 65; CRC is diagnosed more often with advanced age. Being the life expectancy in Spain is around 80 years in males and 83 in females, it could be important to include also older patients.

We have decided to include only patients from 50 to 65 years to achieve a complete follow-up. Patients will be followed for a total of 10 years, which could be a long time to fulfill the follow-up in patients older than 65 years. Remarkably, complete information for most of the patients after this time is important to obtain the necessary data to evaluate the objectives of our trial. We thank the reviewer for demanding clarification of this point. We know that it is a limitation of our study and we have included it in the discussion (lines 303-307).

4-A very detailed Informed Consent From is required to explain the detail of the study to the patients.

We have designed a detailed informed consent that will be sent to the patients. You can find this document attached to this response. The informed consent is written in Spanish due to the patients are recruited only in centers of Spain. Additionally, we have included a sentence in the Ethical and legal aspects section (lines 215-217) to clarify that the informed consent will be obtained from all the patients and that any modification will be correctly communicated.

5-The Authors stated that the general population prefer to have performed FIT rather than colonoscopy. Is this true also for patients who had a diagnosis of advanced adenoma? These patients probably were told about the risk they overcame having the adenoma resected. Are they ready to risk trying a new diagnostic test? I do not have an answer. Do the Author have an answer?

There is a British study that can response to this answer [3]. Bonello B et al included 3100 British individuals without CRC to evaluate the preferences of the population for CRC surveillance. The authors used a hypothetical scenario asking people to imagine they had been diagnosed with intermediate-risk adenomas. Even in this scenario, most of the participants preferred FIT over colonoscopy for surveillance (60.8 % vs. 31.0 %). Moreover, the main reason for preferring FIT was the test is done more frequently. We do not know the preferences for surveillance in Spanish population, but we will analyze this concept using the patients included in our study to have an answer to this question. We have changed a sentence in the discussion to clarify the data published about the preferences for surveillance comes from British population (lines 290-291).

[3] Bonello B et al. Using a hypothetical scenario to assess public preferences for colorectal surveillance following screening-detected, intermediate-risk adenomas: annual home-based stool test vs. triennial colonoscopy. BMC Gastroenterol. 2016 Sep 13;16(1):113.

Reviewer 2 Report

The authors present a paper describing the design of a multicenter and randomized clinical trial within the Spanish population colorectal cancer screening programs to compare fecal immunochemical test (FIT) surveillance with endoscopic surveillance. Patients will be randomly allocated to perform an annual FIT and colonoscopy if fecal hemoglobin concentration is ≥10µg/g, or to perform endoscopic surveillance.

The main endpoints is 10 years colorectal cancer incidence and the secondary endpoints are diagnostic yield, participation, adverse effects, mortality and cost-effectiveness.

Introduction, hypothesis and methodology are well presented and well detailled.

In my opinion, authors only need to correct references according to the instructions for authors format so that the article can be published in our journal

Author Response

We have revised the format of the manuscript to assure compliance with the Diagnostic format. We have modified the references according to the instructions of the journal.